# Factors Predictive of Primary Resistance to Immune Checkpoint Inhibitors in Patients with Advanced Non-Small Cell Lung Cancer

**DOI:** 10.3390/cancers15102733

**Published:** 2023-05-12

**Authors:** Yiqing Huang, Joseph J. Zhao, Yu Yang Soon, Adrian Kee, Sen Hee Tay, Folefac Aminkeng, Yvonne Ang, Alvin S. C. Wong, Lavina D. Bharwani, Boon Cher Goh, Ross A. Soo

**Affiliations:** 1Department of Haematology-Oncology, National University Cancer Institute Singapore, Singapore 119074, Singapore; 2Yong Loo Lin School of Medicine, National University of Singapore, Singapore 117545, Singapore; 3Department of Radiation Oncology, National University Cancer Institute Singapore, Singapore 119074, Singapore; 4Division of Respiratory and Critical Care Medicine, Department of Medicine, National University Hospital, Singapore 119074, Singapore; 5Division of Rheumatology, Department of Medicine, National University Hospital, Singapore 119074, Singapore; 6Department of Biomedical Informatics (DBMI), Yong Loo Lin School of Medicine, National University of Singapore, Singapore 117545, Singapore; 7Department of Medicine, Yong Loo Lin School of Medicine, National University of Singapore, Singapore 117545, Singapore; 8Department of Oncology, Tan Tock Seng Hospital Singapore, Singapore 308433, Singapore

**Keywords:** primary resistance, immune checkpoint inhibitor, non-small-cell lung cancer

## Abstract

**Simple Summary:**

According to the Society for Immunotherapy of Cancer, primary resistance to immune checkpoint inhibitor (ICI) treatment is defined as progression of disease within 6 months of ICI treatment with patients receiving at least 6 weeks of ICI monotherapy. We evaluated factors predictive of primary resistance to ICI monotherapy in 108 advanced non-small-cell lung cancer patients. The prevalence of primary resistance was 54.6%. The majority of patients were male, smokers, received pembrolizumab and had adenocarcinoma histology. We found that female gender, an elevated neutrophil-to-lymphocyte ratio of ≥3 at 6 weeks and a later line of immunotherapy treatment (≥2 lines) were key factors in predicting primary resistance to ICI monotherapy in advanced NSCLC.

**Abstract:**

Introduction: Primary resistance to immune checkpoint inhibitors (ICI) is observed in routine clinical practice. We sought to determine factors predictive of primary resistance to ICI monotherapy, defined by the Society for Immunotherapy of Cancer (SITC) as progression within 6 months of ICI treatment with patients receiving at least 6 weeks of ICI monotherapy, in patients with advanced non-small-cell lung cancer (NSCLC). Method: Patients with stage IV NSCLC treated with at least 6 weeks of single-agent ICI at two tertiary hospitals in Singapore were included. A multivariate logistic regression model was utilised to elucidate factors predictive of primary resistance to ICI. Results: Of the 108 eligible patients, 59 (54.6%) experienced primary resistance. The majority were male (65.7%), smokers (66.3%), Chinese (79.6%), had adenocarcinoma (76.9%), received Pembrolizumab (55.6%) and received immunotherapy treatment in the later line setting (≥2 lines) (61.1%). Female gender (aOR = 3.16, *p* = 0.041), a sixth-week neutrophil-to-lymphocyte ratio (NLR) of ≥3) (aOR = 3.454, *p* = 0.037) and a later line of immunotherapy treatment (≥2 lines) (aOR = 2.676, *p* = 0.040) were factors predictive of primary resistance to ICI monotherapy in patients with advanced NSCLC. Conclusions: Using SITC criteria, an elevated NLR (≥3) at 6 weeks, female gender and a later line of immunotherapy treatment (≥2 lines) were predictive factors of developing primary resistance to ICI monotherapy in patients with advanced NSCLC.

## 1. Introduction

The field of immune checkpoint inhibitors (ICIs) has evolved rapidly in the last decade for lung cancer. In 2016, Reck and colleagues first reported the superiority of pembrolizumab versus platinum-doublet chemotherapy in treatment-naïve metastatic NSCLC with a programmed death ligand-1 (PDL-1) tumour proportion score (TPS) of ≥50%, with an improved objective response rate (ORR), improved overall survival (OS) and progression-free survival (PFS) being found [1,2]. More recently, atezolizumab and cemiplimab both showed an OS and PFS benefit compared with chemotherapy in treatment-naïve advanced NSCLC with high PD-L1 expression [3,4]. The safety profile favoured ICI compared to platinum-doublet chemotherapy, with improved quality of life and a delay in the deterioration of symptoms being found [5]. However, despite the initial impressive results from these studies, disease progression as the best response remains a major clinical problem in patients with advanced NSCLC, with a reported frequency of 21–27% amongst advanced NSCLC with first-line ICI [1,6,7], 7–18% amongst those with first-line ICI in combination with chemotherapy [8,9,10,11], and 20–44% in the pre-treated setting with ICI monotherapy [12,13,14,15,16].

Clinical definitions of primary resistance to ICI lack consistency across the field. An earlier study has defined it as disease progression using RECIST criteria upon the first CT evaluation or death prior to the first CT evaluation [17], whilst another paper defined it as a failure to ever respond [18]. The Immunotherapy Resistance Taskforce was formed by the Society for Immunotherapy of Cancer (SITC) to develop a consensus on the definition of resistance to ICIs. Primary resistance, as established by the SITC, is defined as progression within 6 months of ICI therapy. Patients must have received at least 6 weeks of ICI monotherapy with the best response of progressive disease or stable disease [19].

Using the above definition, we performed a retrospective analysis of advanced NSCLC patients who received ICI monotherapy across two tertiary institutions in Singapore to identify factors predictive of primary resistance to ICI. While the molecular mechanisms of resistance to ICIs have been studied and described extensively [12,20,21], studies on clinical factors predictive of primary resistance are lacking. In this study, we included advanced NSCLC patients on ICI monotherapy only to minimize treatment heterogeneity in the study population, and remained consistent with the SITC definition of primary resistance, where only patients treated with systemic anti-PD1 or anti-PD-L1 monotherapy were included [19].

## 2. Methodology

### 2.1. Study Design and Participants

Patients with stage IV non-small-cell lung cancer, treated with at least 6 weeks of a single-agent immune checkpoint inhibitor at National University Cancer Institute Singapore and Tan Tock Seng Hospital were included and retrospectively analysed. We defined progression of disease using RECIST criteria based on the first surveillance scan. The study was approved by the National Healthcare Group Domain Specific Review Board (reference number: 2017/01254). Informed consent was obtained from all patients who were alive whereas a waiver of consent was obtained for patients who had deceased prior to 29 February 2020. The authors declare that the study was conducted in accordance with the Helsinki Declaration as revised in 2013. Primary resistance was defined as progression within 6 months of ICI therapy. Patients must have had progressive disease or stable disease as the best response per the SITC definition [19]. Patients with a dual ICI combination, ICI–chemotherapy combination and those who discontinued treatment early due to toxicities were excluded.

### 2.2. Data Collection

Demographic, clinical, laboratory, treatment and outcome data were extracted from the electronic health records using a standardised data collection form. Demographic data including age, gender, ethnicity and clinical data, such as Eastern Cooperative Oncology Group (ECOG) performance status, body mass index (BMI), histology, EGFR mutation status, PD-L1 tumour proportion score (TPS), smoking status, and presence of brain metastases, were collected. Treatment data on the type, dose, duration and number of cycles of ICI, the line of treatment and other cancer treatment such as chemotherapy, radiotherapy or targeted therapy prior to or after ICI monotherapy were also collected. Laboratory data on the neutrophil-to-lymphocyte ratio (NLR) and platelet-to-lymphocyte ratio (PLR) at the baseline and at the 6th week were calculated using the formula absolute neutrophil count/absolute lymphocyte count and platelet count/absolute lymphocyte count, respectively. NLR data were analysed as a continuous variable or dichotomised into prespecified cut-offs for ≥3 or <3 [22]. PLR data were analysed as a continuous variable or dichotomised into prespecified cut-offs of ≥180 or <180 [22]. Outcome data of the objective response rate and overall survival were analysed. The objective response rate was assessed, per the criteria used by the SITC taskforce, the RECIST 1.1 criteria [19]. While immune RECIST (iRECIST) has been specifically developed to address issues of mixed responses or pseudo-progression while on immune checkpoint inhibitors, it requires additional validation in assessing the efficacy of anti-PD(L)1 therapy in registration trials [23]. Overall survival was defined as the time of ICI initiation to death from any cause.

### 2.3. Outcomes

The primary outcome is the prevalence of primary resistance to immune checkpoint inhibitors and factors predictive of primary resistance. The secondary outcome includes OS. Pre-identified variables of interest include age, gender, ECOG performance status, ethnicity, BMI, histology type, line of treatment, epidermal growth factor receptor (EGFR) mutation status, PD-L1 TPS, brain metastases, baseline and 6th week NLR and PLR and the presence of primary resistance.

### 2.4. Statistical Analysis

Baseline differences between patients were compared using the Mann–Whitney U test for continuous variables and Fisher’s exact test or Pearson’s χ^2^ test for categorical variables.

A multivariable logistic regression model was utilised to determine the risk factors predictive of primary resistance to ICI. Variables with a *p*-value of <0.1 upon univariable logistic regression and with less than 10% missing data were included in the multivariable logistic regression model. The strength of fit (discrimination) and goodness-of-fit (calibration) of the logistic model were assessed using methods described by Lemeshow and Hosmer and with confidence intervals derived from bootstrap validation.

Overall survival was defined as the start date of ICI to the date of death from any cause. The difference in overall survival between patients with primary resistance and no primary resistance was analysed using the Kaplan–Meier method, log-rank test and univariable Cox proportional hazard regression model. Follow-up time was evaluated using the reverse Kaplan–Meier method.

All analyses were conducted in R-4.1.0, and a two-sided *p*-value of <0.05 was considered statistically significant.

## 3. Results

Between June 2014 to October 2021, we recruited 222 advanced NSCLC patients treated with immune checkpoint inhibitors at the National University Cancer Institute Singapore and Tan Tock Seng Hospital. Sixty-seven patients discontinued treatment early (<6 weeks) due to adverse events. Forty-three patients received the immunotherapy and chemotherapy combination, while 4 patients had dual immune checkpoint inhibitors. There were one hundred and eight patients who were treated with ICI monotherapy for at least 6 weeks and these patients were included for analysis. All patients did not have a prior immune checkpoint inhibitor in their treatment course. Amongst them, 59 (54.6%) encountered primary resistance (Figure 1).

The baseline characteristics of 108 patients included are summarised in Table 1. The median age was 64. Notably, the majority of patients were male (*n* = 71 [65.7%]), Chinese (*n* = 86 [79.6%]), smokers (*n* = 69 [66.3%]), had an ECOG performance status of ≥1 (*n* = 50 [65.8%]) had an adenocarcinoma histology (*n* = 83 [76.9%]), were treated with pembrolizumab (*n* = 60 [55.6%]), and had received immunotherapy treatment in the later line setting (≥2 lines) (*n* = 66 [61.1%]) (Table 1). Amongst those with primary resistance, 76.3% (*n* = 45) had progression of disease as the best response, whilst 23.7% (*n* = 14) had stable disease of less than a 6-month duration. Compared to patients without primary resistance, patients with primary resistance had a higher frequency of progressive disease as the best response (*p* < 0.001).

Compared to patients without primary resistance, more females (*p* = 0.018), patients treated at later lines (*p* = 0.034) and patients with *EGFR*+ tumours (*p* = 0.018) encountered primary resistance. Patients with primary resistance were also noted to have a significantly lower sixth-week lymphocyte count, a higher sixth-week neutrophil-to-lymphocyte ratio (NLR) when dichotomised at 3 and 5, and a sixth-week platelet to lymphocyte ratio (PLR) when dichotomised at 180 (Table 1).

Upon the univariable logistic regression, female gender (*p* = 0.020), line of treatment (*p* = 0.007), the presence of *EGFR*+ tumours (*p* = 0.019), pre-treatment lymphocyte count (*p* = 0.017), sixth-week lymphocyte count (*p* = 0.017), and sixth-week NLRs of ≥3 (*p* = 0.003) and ≥5 (*p* = 0.028) were significantly associated with development of primary resistance (Table 2).

The multivariable logistic regression model exhibited acceptable discrimination (AUC = 0.749, 95% confidence interval (CI) = 0.643–0.84 computed with 2000 stratified bootstrap replicates; Figure 2). There was no evidence to reject the null hypothesis that the model fit the data (Hosmer–Lemeshow test: χ^2^ = 1.309, *p* = 0.995) (Figure 2). Upon the multivariable logistic regression, female gender (aOR = 3.165, 95% CI = 1.078–10.101, *p* = 0.041), a sixth-week NLR of ≥ 3 (aOR = 3.454, 95% CI = 1.102–11.643, *p* = 0.037), and a later line of immunotherapy treatment (≥2 lines) (aOR = 2.676, 95% CI = 1.056–7.008, *p* = 0.040) remained predictive factors of primary resistance to ICI monotherapy in patients with advanced NSCLC. While lymphocyte counts at pre-treatment and at the sixth week were both significantly associated with primary resistance upon univariate analysis, only the sixth-week lymphocyte count was included in the multivariate logistic regression model in view of a larger magnitude of treatment effect. EGFR mutation was also not included in the multivariate analysis in view of the high proportion of missing data, where 29 patients had an unknown EGFR mutation status.

The median duration of ICI therapy was 2.7 months (82 days) for patients with primary resistance compared to 11.7 months (352 days) for those without primary resistance (*p* < 0.001). The median duration of follow-up was 49.2 months. Patients with primary resistance had a significantly shorter survival duration compared to those without primary resistance (unadjusted HR = 0.264, 95% CI = 0.163–0.427, *p* < 0.0001; median survival time = 9.76 vs. 19.33 months) (Appendix A).

## 4. Discussion

Reports on the clinical characteristics of primary resistance to ICIs among NSCLC patients have been sparse. In a study of 93 pre-treated advanced NSCLC patients in the United States who received ICI monotherapy, never-smokers or those who smoked fewer pack years, more involved metastatic sites, more prior therapies and a lower mean albumin level were reported as factors predictive of primary resistance to ICI therapy [17]. The authors reported a primary resistance in 38.7% of patients and defined primary resistance as progressive disease upon the first radiological evaluation or death prior to the first CT evaluation [17]. A retrospective study in France conducted by Bernichon and colleagues involving 96 NSCLC patients treated with nivolumab did not identify any clinical factors that were predictive of primary or secondary resistance [24]. Factors associated with the response to ICIs have been described amongst patients with advanced NSCLC. An exploratory analysis of 268 NSCLC patients treated with anti-PD-1 therapy at the Princess Margaret Cancer Centre in Canada found that current and former smokers had significantly higher response rates compared to non-smokers [25]. Similarly, the increase in smoking years was also associated with positive anti-PD-1 therapy response [26]. A high Patras Immunotherapy Score (PIOS) (calculated using the formula performance status x body mass index/lines of treatment x age) was also reported to be associated with better response with anti-PD-1 treatment in advanced NSCLC [27]. Our study is one of the largest retrospective studies involving 108 advanced NSCLC patients treated in Asia. We had a homogenous population, with all patients being treated with anti-PD1 or anti-PD-L1 monotherapy. In addition, we adhered to a standardised definition of primary resistance per the SITC, a key feature absent in earlier studies. While there have been several definitions of primary resistance, we chose the SITC definition as it was generated by a multistakeholder taskforce comprising experts in cancer immunotherapy from academia, industry and the US government, with the goal to provide guidance for clinical trial design and analyses surrounding mechanisms of resistance to immune checkpoint inhibitors [19]. To our knowledge, this is the first study evaluating ICI resistance using the SITC definition. In our study, clinical factors were studied in detail, along with peripheral blood markers. We found that female gender, elevated NLR at six weeks and line of treatment were key factors in predicting primary resistance to ICI.

Progression of disease as the best response has been reported at a frequency of 21–27% amongst advanced NSCLC upon first-line ICI monotherapy [1,6,7] and at a frequency of 20–44% in the second-line setting with ICI monotherapy [12,13,14,15,16]. In our study, the frequencies of primary resistance amongst those treated in the first- and second-line settings were 27.1% and 32.2%, respectively—a finding similar to that reported in the literature. Overall, more than half (54.6%) of the patients encountered primary resistance. This frequency is higher than that reported in the literature. One reason that could explain this is the relatively high proportion of heavily pre-treated patients in our cohort. Approximately 30% of patients had received ICI in the third line setting and beyond for their advanced cancer.

Female gender predicted for primary resistance to ICI monotherapy in NSCLC patients in our study. Whether or not gender difference confers a survival difference to ICI therapy remains unclear. Some studies report that females have an increased survival benefit [28], while others have supported favourable outcomes in men compared to women [29,30]. A meta-analysis by Conforti and colleagues involving 1672 advanced NSCLC patients found that anti-PD1 or anti-PD-L1 monotherapy was highly effective in men but not in women, even in patients expressing high PD-L1 levels [31]. A recent study of 9000 NSCLC patients, however, found that ICI confers a similar survival benefit regardless of gender [32]. In contrast, a pooled analysis reported that women with advanced NSCLC derived a larger benefit from the addition of chemotherapy to anti-PD-1/anti-PD-L1 compared to men [33]. From the biological perspective, several mechanisms have been postulated to explain the gender differences. Firstly, immune responses between men and women differ, with women exhibiting a higher efficiency of antigen-presenting cells and macrophage activation, and higher levels of B cells, antibody production, CD4+ T-cells and T helper 2 cell response, while men expressed higher levels of CD8+ T-cells, regulatory T-cells and Th1 cell response [34]. Secondly, immune response can be modulated by hormones including oestrogen, progesterone and testosterone [35,36]. Gender differences in gut microbiome composition may also impact immune competency [37]. Lastly, the difference in expression of X-linked immune-related genes such as TLR7, TLR9, IL-2, IL-4 and IL-15 has been reported to drive the difference in response to ICI between the two genders [38]. As there is a lower representation of women in clinical trials [32], future studies involving a larger number of female patients can help shed light on the impact of gender as a predictor of ICI response in NSCLC.

Elevated NLR at the sixth week was also a predictor of primary resistance to ICI monotherapy in advanced NSCLC. Systemic inflammation plays a crucial role in tumour development and has been associated with prognosis in solid tumours due to its effect on the immune response to the disease [39,40]. Neutrophil-to-lymphocyte ratio (NLR) has been described as a marker of the general immune response to stress stimuli [41,42]. Earlier studies have reported that an elevated neutrophil-to-lymphocyte ratio (NLR) was associated with poorer outcomes in NSCLC treated with ICIs [22,43,44,45,46]. In a multi-centre retrospective study of 466 NSCLC patients across Europe, Mezquita and colleagues reported that a pre-treatment NLR of > 3 was correlated with a worse outcome for ICI, but not for chemotherapy [46]. Similarly, a meta-analysis of 21 studies involving 1845 NSCLC patients demonstrated that a high pre-treatment NLR and platelet-to-lymphocyte ratio (PLR) were associated with poorer outcomes in patients treated with ICIs [47]. In our study, whilst baseline NLR was not a predictive factor, elevated NLR at the sixth week was predictive of primary resistance to ICI therapy. This finding is consistent with earlier reports. A retrospective study in Korea found that a high post-treatment NLR of ≥5 was associated with poor prognosis in advanced NSCLC patients receiving an anti-PD1 inhibitor [48]. Similarly, a retrospective study of 41 small-cell lung cancer patients in China found that a post treatment NLR of ≥5 was associated with a shorter PFS with ICI therapy in the second line or later-line setting [49]. Taken together, NLR is a simple blood-based biomarker that can help predict resistance to ICI therapy amongst lung cancer patients. Future prospective studies are warranted to validate its use in the clinical setting.

In this study, lower pre-treatment and sixth-week lymphocyte counts were significantly associated with a risk of primary resistance to ICI upon univariate analysis. The sixth-week lymphocyte count was eventually included in the multivariate logistic regression model in view of a larger magnitude of treatment effect. An earlier retrospective study conducted at Johns Hopkins hospital demonstrated that a lower absolute pre-treatment lymphocyte count was associated with less clinical benefits with anti-PD-1 therapy in recurrent or metastatic head and neck squamous cell carcinomas [50]. Although the sixth-week lymphocyte count did not reach statistical significance upon multivariate logistic regression analysis in our study, both pre-treatment and sixth-week lymphocyte count remain potential biomarkers that can be explored in future immunotherapy trials.

Thirty-nine percent of patients received ICI monotherapy in the first line in this study, while the rest of the patients were treated in the second-line setting and beyond. Notably, 32.4% of the patients received ICI in the third-line setting and beyond. As this study recruited patients treated with ICI monotherapy between June 2014 and October 2021, and many of the heavily pre-treated patients were recruited before ICI was established as a standard first- or second-line treatment in advanced NSCLC. We reported that a later line of immunotherapy treatment (≥2 lines) was one of the key predictors of primary resistance to ICIs. This finding is consistent with that of prior studies, showing poorer response rates in subsequent lines of therapies in advanced NSCLC. An earlier study on advanced NSCLC using ICI monotherapy also confirmed the same finding [17].

We acknowledge several limitations. Firstly, this was a retrospective study conducted across two institutions. Secondly, the sample size was relatively small at 108 patients. Thirdly, the population studied was heterogeneous; we included patients who received front-line and later-line immunotherapy treatment, patients of all PD-L1 TPS were studied and 19 patients carried an *EGFR* mutation. Fourthly, while the SITC required a confirmatory CT scan 4 weeks after the initial progression of the disease, none of our patients had a confirmatory second CT scan. Progression of disease was assessed at the first radiological assessment. Lastly, this study also did not include coupling of genomic data in predicting primary resistance to ICI in our patients. Nonetheless, to our knowledge, this is the first study reporting factors predictive of primary resistance among advanced NSCLC patients, while adhering to a uniform SITC definition of primary resistance.

Mechanisms behind primary resistance to ICI have been extensively studied, spanning from tumour factors (intrinsic and extrinsic factors) to host factors [12,20,21,51]. Tumour-intrinsic mechanisms include a lack of tumour immunogenicity (low tumour mutational burden (TMB), heterogeneous antigens, and mutation of certain genes), loss of tumour antigen expression, loss of HLA expression, aberration in signalling pathways such as those of mitogen-activated protein kinase (MAPK), PI3K, WNT, and IFN, and constitutive PD-L1 expression [21,52,53,54,55]. Extrinsic factors, on the other hand, involve components other than tumour cells within the tumour microenvironment. These include the presence of regulatory T-cells which suppress the effector T-cell response [56,57], myeloid-derived suppressor cells which are implicated in promoting tumour angiogenesis, cell invasion and metastases [58] and tumour-associated M2 macrophages which can affect the response to ICI therapy [59,60]. Other extrinsic factors that have been described include T-cell related factors (alternative immune checkpoints, T-cell exhaustion and phenotype alteration, T-cell receptor (TCR) repertoire, and epigenetic modification), cytokines and metabolites (e.g., TGF-B, adenosine) related to the tumour micro-environment [21,61,62]. Host-related characteristics leading to primary resistance including alterations in the gut microbiome, antibiotic use, inflammation state and autoimmunity have also been described [63,64].

Currently, to reduce the rate of primary resistance to ICI, therapeutic strategies have included the addition of chemotherapy to ICI therapy upfront [8,9] or, upon ICI progression (ClinicalTrials.gov identifier: NCT03793179), combining dual ICIs with or without chemotherapy [65,66], and adding novel agents or targeted therapies to ICI [12]. To enhance precision medicine and personalised cancer immunotherapy, advances in biomarker development are currently ongoing together with efforts in understanding the resistance mechanisms against ICIs.

In conclusion, in this retrospective study of 108 advanced NSCLC patients receiving ICI monotherapy, an elevated NLR at the sixth week, female gender and a later line of immunotherapy treatment (≥2 lines) were predictors of primary resistance to ICI therapy. Future larger studies are warranted to validate our key findings.

## Figures and Tables

**Figure 1 cancers-15-02733-f001:**
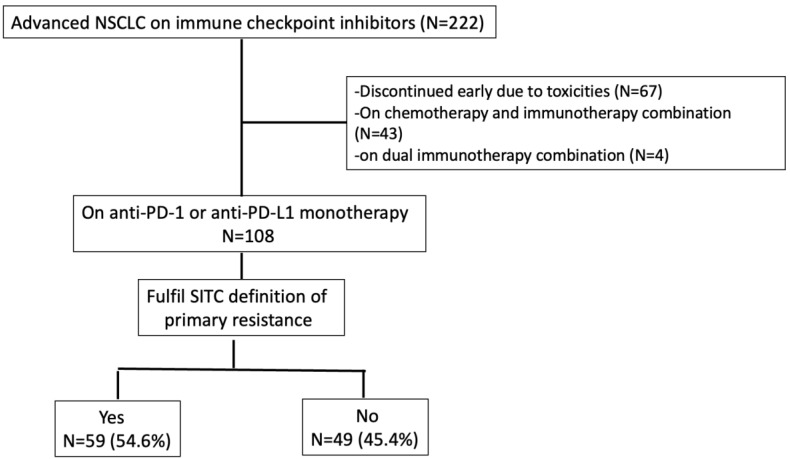
Consort diagram.

**Figure 2 cancers-15-02733-f002:**
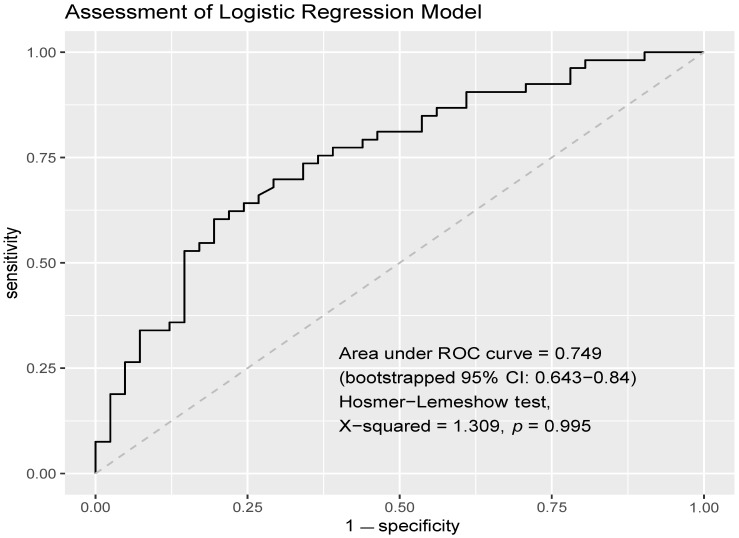
AUROC of multivariate logistic regression model. The multivariable logistic regression model exhibited acceptable discrimination (AUC = 0.749, 95% CI = 0.643–0.84). There was no evidence to reject the null hypothesis that the model fits the data (Hosmer–Lemeshow test: χ^2^ = 1.309, *p* = 0.995).

**Table 1 cancers-15-02733-t001:** Patient demographics.

		Overall	No Primary Resistance	Primary Resistance	*p*
N		108	49	59	
Age (median (IQR))		64.17 (5.44, 71.50)	66.25 (57.89, 72.82)	62.00 (53.64, 70.13)	0.144
Gender (%)	Female	37 (34.3)	11 (22.4)	26 (44.1)	0.018
	Male	71 (65.7)	38 (77.6)	33 (55.9)	
Ethnicity (%)	Chinese	86 (79.6)	39 (79.6)	47 (79.7)	0.685
	Indian	4 (3.7)	3 (6.1)	1 (1.7)	
	Malay	13 (12.0)	5 (10.2)	8 (13.6)	
	Others	5 (4.6)	2 (4.1)	3 (5.1)	
Smoking status (%)	No	35 (33.7)	12 (24.5)	23 (41.8)	0.062
	Yes	69 (66.3)	37 (75.5)	32 (58.2)	
Performance status (%)	0	26 (34.2)	13 (38.2)	13 (31.0)	0.506
	≥1	50 (65.8)	21 (61.8)	29 (69.0)	
Line of treatment (%)	1	42 (38.9)	26 (53.1)	16 (27.1)	0.034
	2	31 (28.7)	12 (24.5)	19 (32.2)	
	3	17 (15.7)	4 (8.2)	13 (22.0)	
	≥4	18 (16.7)	7 (14.3)	11 (18.6)	
Brain metastasis (%)	No	78 (72.2)	39 (79.6)	39 (66.1)	0.119
	Yes	30 (27.8)	10 (20.4)	20 (33.9)	
Histology (%)	Adenocarcinoma	83 (76.9)	39 (79.6)	44 (74.6)	0.536
	Squamous cell carcinoma	13 (12.0)	4 (8.2)	9 (15.3)	
	Others	12 (11.1)	6 (12.2)	6 (10.2)	
PDL1 level (%)	<1%	5 (8.8)	3 (11.1)	2 (6.7)	0.916
	1–49%	13 (22.8)	6 (22.2)	7 (23.3)	
	≥50%	39 (68.4)	18 (66.7)	21 (70.0)	
EGFR mutation (%)	No	60 (75.9)	32 (88.9)	28 (65.1)	0.018
	Yes	19 (24.1)	4 (11.1)	15 (34.9)	
BMI (kg/m^2^) (median [IQR])		21.10 [19.10, 24.16]	21.12 [20.06, 24.07]	21.00 [19.00, 24.17]	0.641
Best response (%)	CR	1 (0.9)	1 (2.0)	0 (0.0)	<0.001
	PD	49 (45.4)	4 (8.2)	45 (76.3)	
	PR	23 (21.3)	23 (46.9)	0 (0.0)	
	SD	35 (32.4)	21 (42.9)	14 (23.7)	
Immunotherapy agent (%)	Pembrolizumab	60 (55.6)	28 (57.1)	32 (54.2)	
	Nivolumab	24 (22.2)	7 (14.3)	17 (28.8)	
	Durvalumab	11 (10.2)	5 (10.2)	6 (10.2)	
	Atezolizumab	8 (7.4)	5 (10.2)	3 (5.1)	0.203
	Avelumab	5 (4.6)	4 (8.2)	1 (1.7)	
6th week lymphocyte count (×10^9^ L) (median [IQR])		1.21 [0.94, 1.65]	1.36 [1.09, 2.02]	1.14 [0.83, 1.50]	0.007
Pre-treatment NLR (%)	<3	19 (29.2)	10 (40.0)	9 (22.5)	0.131
	≥3	46 (70.8)	15 (60.0)	31 (77.5)	
	<5	36 (55.4)	16 (64.0)	20 (50.0)	0.269
	≥5	29 (44.6)	9 (36.0)	20 (50.0)	
Pre-treatment PLR (%)	<180	33 (35.1)	16 (40.0)	17 (31.5)	0.392
	≥180	61 (64.9)	24 (60.0)	37 (68.5)	
6th week NLR (%)	<3	31 (31.6)	20 (48.8)	11 (19.3)	0.002
	≥3	67 (68.4)	21 (51.2)	46 (80.7)	
	<5	59 (60.2)	30 (73.2)	29 (50.9)	0.026
	≥5	39 (39.8)	11 (26.8)	28 (49.1)	
6th week PLR (%)	<180	27 (29.0)	15 (40.5)	12 (21.4)	0.047
	≥180	66 (71.0)	22 (59.5)	44 (78.6)	

CR: complete response; EGFR: epidermal growth factor receptor; NLR: neutrophil-to-lymphocyte ratio; PD: progressive disease; PDL1: programmed death ligand 1; PLR: platelet-to-lymphocyte ratio; PR: partial response; SD: stable disease.

**Table 2 cancers-15-02733-t002:** Univariate and multivariate logistic regression.

Variable	Comparison	Reference	OR	*p*	Adjusted-OR	*p*
Age			0.979 (0.946–1.012)	0.216		
Gender	Female (*n* = 37)	Male (*n* = 71)	2.725 (1.192–6.536)	0.020	3.165 (1.078–10.101)	0.041
Ethnicity	Indian (*n* = 4)	Chinese (*n* = 86)	0.277 (0.013–2.257)	0.274		
	Malay (*n* = 13)	Chinese (*n* = 86)	1.328 (0.409–4.699)	0.642		
	Others (*n* = 5)	Chinese (*n* = 86)	1.245 (0.197–9.808)	0.816		
Smoking status	Yes (*n* = 69)	No (*n* = 35)	0.451 (0.190–1.034)	0.064	1.046 (0.330–3.389)	0.939
Alcohol	Yes (*n* = 27)	No (*n* = 57)	0.628 (0.247–1.580)	0.323		
Performance status	0 (*n* = 26)	≥1 (*n* = 50)	0.724 (0.277–1.882)	0.506		
Line of treatment	≥2 (*n* = 66)	1 (*n* = 42)	3.040 (1.377–6.897)	0.007	2.676 (1.056–7.008)	0.040
Brain metastasis	Yes (*n* = 30)	No (*n* = 78)	2.000 (0.845–4.972)	0.122		
Histology	Others (*n* = 12)	Adenocarcinoma (*n* = 83)	0.886 (0.257–3.051)	0.845		
	Squamous cell carcinoma (*n* = 13)	Adenocarcinoma (*n* = 83)	1.994 (0.598–7.831)	0.281		
PD-L1 level	≥50% (*n* = 39)	<1% (*n* = 5)	1.750 (0.262–14.424)	0.563		
	1–49% (*n* = 13)	<1% (*n* = 5)	1.750 (0.217–16.980)	0.601		
EGFR mutation	Yes (*n* = 19)	No (*n* = 60)	4.286 (1.371–16.413)	0.019		
BMI (kg/m^2^)			0.995 (0.898–1.102)	0.920		
Pre-treatment lymphocyte count (×10^9^ L)			0.464 (0.237–0.844)	0.017		
6th week lymphocyte count (×10^9^ L)			0.452 (0.222–0.829)	0.017	0.769 (0.321–1.691)	0.528
Pre-treatment NLR ≥ 3	≥3 (*n* = 46)	<3 (*n* = 19)	2.296 (0.773–6.988)	0.135		
Pre-treatment NLR ≥ 5	≥5 (*n* = 29)	<5 (*n* = 36)	1.778 (0.646–5.095)	0.271		
6th week NLR ≥ 3	≥3 (*n* = 67)	<3 (*n* = 31)	3.983 (1.651–10.068)	0.003	3.454 (1.102–11.643)	0.037
6th week NLR ≥ 5	≥5 (*n* = 39)	<5 (*n* = 59)	2.633 (1.130–6.433)	0.028		
Pre-treatment PLR ≥ 180	≥180 (*n* = 61)	<180 (*n* = 33)	1.451 (0.616–3.430)	0.393		
6th week PLR ≥ 180	≥180 (*n* = 66)	<180 (*n* = 27)	2.500 (1.007–6.357)	0.050		

EGFR: epidermal growth factor receptor; NLR: neutrophil-to-lymphocyte ratio; PDL1: programmed death ligand 1; PLR: platelet-to-lymphocyte ratio.

## Data Availability

The data presented in this study are available upon request from the corresponding author.

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
