# Peer review of "Factors Predictive of Primary Resistance to Immune Checkpoint Inhibitors in Patients with Advanced Non-Small Cell Lung Cancer"

_cancers, 2023, doi:10.3390/cancers15102733_

Round 1

Reviewer 1 Report

Huang and colleagues evaluated predictors for primary IO resistance in 108 patients with advanced NSCLC receiving anti-PD(L)1 checkpoint monotherapy as 1L/2L/3L/4L treatment, and found gender, neutrophil-lymphocyte ratio and treatment beyond 1L to be significant predictors of primary IO resistance. The manuscript is well written. There are several limitations of the study, most importantly the very heterogeneous study population (front-line and later-line IO monotherapy, all PDL1 status included, mEGFR status in 24% of patients); although the authors adressed this issue by using multivariate regression modeling, it is important to discuss this limitation in the Discussion Section. The term ‘treatment beyond 1st line’ is confusing, as the authors’ stratification was rather 1L versus later-line systemic IO treamtent. Additionally, the line of IO (palliative?) treatment makes only sense as a potential predictor of IO resistance if patients were IO-naive at the time of study-relevant IO treatment, which is an issue the authors should clarify for the reader. With this regards, it’s also difficult to understand if the authors state that they excluded patients that received IO concurrently with chemotherapy, i.e. does this mean none of the included patients received first-line chemo-IO even though this would be standard treatment for approx. 31% of the study patients (with a PDL1 status <50%)?

Figure 3: It is not adequate to plot OS in patients with advanced NSCLC over all treatment lines.

well written manuscript

Author Response

Thank you very much for your constructive feedback on our manuscript. We have reviewed each comment thoroughly and addressed them accordingly. Please kindly refer to the table below summarizing our replies to the individual comments.

Once again, thank you very much for your kind consideration of our manuscript.

Yours sincerely,

Yiqing Huang

Reviewer 1’s comments

Authors’ reply

Huang and colleagues evaluated predictors for primary IO resistance in 108 patients with advanced NSCLC receiving anti-PD(L)1 checkpoint monotherapy as 1L/2L/3L/≥4L treatment, and found gender, neutrophil-lymphocyte ratio and treatment beyond 1L to be significant predictors of primary IO resistance. The manuscript is well written.

There are several limitations of the study, most importantly the very heterogeneous study population (front-line and later-line IO monotherapy, all PDL1 status included, mEGFR status in 24% of patients); although the authors adressed this issue by using multivariate regression modeling, it is important to discuss this limitation in the Discussion Section.

Thank you very much for this helpful comment. We have highlighted this as a limitation in our Discussion section.

The term ‘treatment beyond 1st line’ is confusing, as the authors’ stratification was rather 1L versus later-line systemic IO treatment.

Thank you very much for this valuable comment. We have amended this phrase to “later line of immunotherapy treatment (≥2 lines)” in our manuscript.

Additionally, the line of IO (palliative?) treatment makes only sense as a potential predictor of IO resistance if patients were IO-naive at the time of study-relevant IO treatment, which is an issue the authors should clarify for the reader.

Thank you for your comment.

We have checked that all our patients were IO-naïve at the time of study-relevant IO treatment. We have indicated a sentence on this in the “Results section” to clarify this to readers as well.

With this regards, it’s also difficult to understand if the authors state that they excluded patients that received IO concurrently with chemotherapy, i.e. does this mean none of the included patients received first-line chemo-IO even though this would be standard treatment for approx. 31% of the study patients (with a PDL1 status <50%)?

Thank you for this valuable comment.

This study only recruited patients on IO-monotherapy. We excluded patients who received chemo-IO combination and those who received dual IO combination. This is stated in the “Methodology” section, under ‘Study Design and Participants’ at the last line.

Figure 3: It is not adequate to plot OS in patients with advanced NSCLC over all treatment lines.

Dear reviewer, we wanted to demonstrate the poorer outcomes of patients who developed primary resistance compared to those who did not. However, we understand that there is limitation in plotting this OS curve in patients across all treatment lines. Hence, we will move this figure to the supplementary section. We have changed it to a supplementary figure.

Reviewer 2 Report

Dear Authors,

please find my few comments on the manuscript below:

1. Head of Table 1: "No primary resistance" instead of "Nil primary resistance"?

2. Lines 155-160: Best response (0%) not mentioned as significantly different between no-resistance and resistance cohorts.

3. Lines 161-164: EGFR mutation is not mentioned as significantly associated with primary resistance. 

4. Tiltel of OX axis in Figure 2: "1-specificity" instead of "specificity".

Best regards

Author Response

Thank you very much for your constructive feedback on our manuscript. We have reviewed each comment thoroughly and addressed them accordingly. Please kindly refer to the table below summarizing our replies to the individual comments.

Once again, thank you very much for your kind consideration of our manuscript.

Yours sincerely,

Yiqing Huang

Reviewer 2’s comments

Authors’ Reply

please find my few comments on the manuscript below:

1.Head of Table 1: "No primary resistance" instead of "Nil primary resistance"?

Dear reviewer, thank you very much for your comments.

Yes, we have changed the head of Table 1 accordingly.

2. Lines 155-160: Best response (0%) not mentioned as significantly different between no-resistance and resistance cohorts.

Thank you for this comment. We have made the necessary changes and described this in the “Results” section.

3. Lines 161-164: EGFR mutation is not mentioned as significantly associated with primary resistance. 

Thank you very much for this comment. We have indicated that EGFR+ tumours were associated with development of primary resistance on univariate analysis in the “Results” section. We did not include EGFR mutation status in the multivariate analysis due to a significant proportion of missing data. We have explained that in the amended manuscript as well.

4. Title of OX axis in Figure 2: "1-specificity" instead of "specificity".

Thank you for this constructive comment. We have amended Figure 2’s X-axis accordingly.

Reviewer 3 Report

 “Factors predictive of Primary Resistance to immune checkpoint 2 inhibitors in Asian patients with advanced non-small cell lung  cancer” is studied in very recent. The contribution of this paper is interesting. I also provide some comments for the author to improve this paper. 

1 Some of the recent related literature is missing. I suggest the author enhance the literature review and review the recent development of predictive Factors to immune checkpoint 2 inhibitors .

2 the title is not suitable in Asian patients, the samples are not from different countries in Asian countries,mostly from Chinese.In addition, please check the title language error.

3  the sample size is small to conclude the result.

4 In Table 1,several factors of baseline  in two group indicate statistic significant difference, please explain it.

5 p value of Pre-treatment lymphocyte count (x10^9L) is 0.017,please discuss its valuable clinical significance  in the discussion section.

6 the p value result from Table 2. Univariate and multivariate logistic regression included gender factor,please explain why they have gender difference ?

7 what did fig 2 and 3 indicate in the result section? the legends are too short,please add more information.

Author Response

Thank you very much for your constructive feedback on our manuscript. We have reviewed each comment thoroughly and addressed them accordingly. Please kindly refer to the table below summarizing our replies to the individual comments.

Once again, thank you very much for your kind consideration of our manuscript.

Yours sincerely,

Yiqing Huang

Reviewer 3’s comments

Authors’ reply

“Factors predictive of Primary Resistance to immune checkpoint 2 inhibitors in Asian patients with advanced non-small cell lung  cancer” is studied in very recent. The contribution of this paper is interesting. I also provide some comments for the author to improve this paper. 

1 Some of the recent related literature is missing. I suggest the author enhance the literature review and review the recent development of predictive Factors to immune checkpoint 2 inhibitors .

Dear reviewer, thank you very much for this feedback. We have enhanced our literature review, especially in terms of clinical factors predictive of resistance or response to ICIs and have made the necessary changes in the 1st paragraph of our “Discussion” section. The molecular mechanism of resistance to ICIs is also described in the later segment of our “Discussion” section.

2 the title is not suitable in Asian patients, the samples are not from different countries in Asian countries,mostly from Chinese.In addition, please check the title language error.

Thank you for this constructive comment. We have decided to remove “Asian” from our title and have made the necessary amendment.

3  the sample size is small to conclude the result.

We acknowledged this limitation of a relatively small sample size of N=108 and have indicated it as a limitation in our discussion section.

4 In Table 1,several factors of baseline  in two group indicate statistic significant difference, please explain it.

Dear reviewer, thank you for this comment. We have described the difference in some of the baseline factors in Table 1 between patients with primary resistance and those with no primary resistance in the “Results” section. We have also expanded on the significant predictive factors found on multivariate analysis and discussed them in our “Discussion” section, in particular female gender, 6thweek NLR and later line of immunotherapy treatment. We hope the amended manuscript is acceptable to you and your team.

5 p value of Pre-treatment lymphocyte count (x10^9L) is 0.017,please discuss its valuable clinical significance  in the discussion section.

Thank you very much for this comment. We have added a segment on pre-treatment lymphocyte count in our “Discussion” section.

6 the p value result from Table 2. Univariate and multivariate logistic regression included gender factor,please explain why they have gender difference ?

Dear reviewer, thank you very much for your kind comment. Female gender is one of the predictors of primary resistance to immune checkpoint inhibitor in our study. We have discussed gender difference and its impact on response to immunotherapy in the “Discussion” section.

7 what did fig 2 and 3 indicate in the result section? the legends are too short,please add more information.

Thank you for your comments. We have added more information to the figure legends for both figures.

Figure 2 will remain, while Figure 3 will be changed to Supplementary Figure 1 in the revised manuscript.

Round 2

Reviewer 1 Report

The authors adressed all issues raised by the reviewer.

Reviewer 3 Report

accept